# Investigation of Channel Mobility Enhancement Techniques Using Si/SiGe/GeSn Materials in Orthogonally Oriented Selective Buried Triple Gate Vertical Power MOSFET: Design and Performance Analysis

**DOI:** 10.3390/mi16040452

**Published:** 2025-04-11

**Authors:** M. Ejaz Aslam Lodhi, Abdul Quaiyum Ansari, Sajad A. Loan, Shabana Urooj, Nidal Nasser

**Affiliations:** 1Department of Electronics and Communication Engineering, Indira Gandhi Delhi Technical University for Women, Delhi 110006, India; ejaz.iitk@gmail.com; 2Department of Electrical Engineering, Jamia Millia Islamia, New Delhi 110025, India; aqansari@jmi.ac.in; 3Department of Electronics and Communication Engineering, Jamia Millia Islamia, New Delhi 110025, India; sloan@jmi.ac.in; 4Department of Electrical Engineering, College of Engineering, Princess Nourah bint Abdulrahman University, P.O. Box 84428, Riyadh 11671, Saudi Arabia; 5College of Engineering, Alfaisal University, Riyadh 11533, Saudi Arabia; nnasser@alfaisal.edu

**Keywords:** triple gate, trench, selective buried, silicon–germanium, breakdown voltage, on-resistance, germanium–tin, mobility, power MOSFET

## Abstract

The performance of the Si MOSFET is suppressed when the channel loses its control through the gate. This paper introduces a new and novel high-channel conducting orthogonally oriented selective buried triple gate vertical power MOSFET technology to study the channel behavior compared with the conventional Si power MOSFET. Our paper investigates the performance of the proposed selective buried triple gate power MOSFET by using different channel materials (SiGe/GeSn over Si) to compare with the conventional Si MOSFET. Our 2D Silvaco simulation output significantly improves device on-current, ON-resistance, channel electron mobility, transconductance, and enhancement in various parameters governing power MOSFET. The unique design of our proposed triple gate gives very high channel mobility of 880 cm^2^/V·s, which we believe to be significant in the triple gate power MOSFET domain. The results show that our optimized triple-gate device achieves an ultra-low specific ON-resistance of 0.31 mΩ·cm², improving Balliga’s FOM1 by 411.61% and FOM2 by 98.704%. This makes it suitable for high-speed and switching devices, compatible with various high-mobility channel materials, and well-suited for future CMOS applications.

## 1. Introduction

Research on keeping the device source and drain terminals in the same horizontal line for older power devices that relied on lateral orientation ended up having very high specific on-resistance [1], mostly because of the device’s large active area [2]. A study on the available state-of-the-art vertical power MOSFET reduced the aforementioned issue. Additionally, prior studies on semiconductor power devices have effectively shown that a double gate is superior to a single gate in overcoming the instability of Vth brought on by fluctuations in channel length [3]. However, the device suffered from sudden changes in the electric field between the two gates [4], whereas single trench gate conventional MOSFET suffers from severe crowding of the electric field at the corner of the trench gate, which degrades the breakdown voltage of the device.

Hence, more studies are required on power MOSFETs to enhance channel control and tradeoffs for superior device performance [5]. Trench-etched double-diffused MOS (TED MOS) technology [6] and triple gate nanowire transistors are two examples of triple gate devices that have recently demonstrated a new dimension in the continuing research because of their exceptional gate control and higher device on-current over planar devices. Using a FinFET construction with numerous sub-micron channels for high device current, Ramamurthy et al. [7] investigated triple gate MOSFETs. However, the most important and concerning characteristic is the MOSFET device ON-current, which is a direct consequence of improved gate-to-channel coupling.

Furthermore, for more than 30 years, silicon has dominated the semiconductor industry. Although the first transistor was built of Ge [8], other semiconductor materials may have larger bandgaps, higher mobilities, or faster saturation velocities, but silicon devices still make up more than 95% of all microelectronics [8]. As the typical device scaling approaches its physical limit, it has become more challenging to improve the performance of Si MOSFETs in recent years [3]. To further enhance CMOS performance, new semiconductor materials are sought in the pursuit of high-mobility transistors. Alternative channel materials are still required for higher mobility MOSFETs, even if Ge MOSFETs exhibit promising qualities to reach an improved performance above Si MOSFETs. One potential option for upcoming high-mobility MOSFETs, particularly for high-mobility, is the use of Si with germanium (Ge) and tin (Sn). Research has clearly shown that higher channel mobility is effective in improving the performance of MOSFETs [8]. Hence, Ge and GeSn, over Si, are interesting choices among these high-mobility materials [9]. Using Si with Ge technology on Si wafers introduced bandgap and strain engineering to improve some of silicon’s lackluster qualities while maintaining the established and affordable fabrication procedures. Since the SiGe MOSFET was first introduced, the market for SiGe devices in the RF industry has expanded. It is anticipated that a large portion of this growth will occur in RF sectors where more expensive III-V materials have historically predominated.

One of the applications in MOSFET is the GeSn channel MOSFET. Due to its excellent carrier mobility and fine suitability with Si VLSI technology, GeSn is a viable channel material for high-performance complementary metal-oxide-semiconductor devices. Despite compressive strain being used to obtain a high hole mobility in GeSn [9], a high electron mobility necessitates a high enough Sn composition in GeSn to support more electrons with reduced effective masses in the valley [9].

This article introduces a very new and novel high-channel conducting Selective Buried triple gate vertical power MOSFET. The present work aims to enhance the channel electron mobility in the device, which is the essential physics of any MOSFET. Our proposed device consists of two selective buried lateral gates and one vertical trench into the drift region. The proposed device has multiple inversion layers due to two lateral gates and one vertical trench gate, which results in excellent gate engineering. Our proposed device with orthogonally oriented gates and the use of a GeSn material results in high device electron mobility and transconductance.

Our proposed device’s novelties and uniqueness are highlighted pointwise. Firstly, the three trench gates are orthogonally oriented to form a vertical triple gate power MOSFET, presented here for the first time. Secondly, the proposed device consists of multiple inversion channels formed in the two p-body regions above the two lateral gates and two sidewalls of the vertical trench gate, simultaneously. Thirdly, all the channels contribute simultaneously to on-state current distribution from the top of the device to the bottom of the drift region, at the drain, resulting in very high on-current, thereby reducing the specific on-resistance (Ron.sp). Fourthly, selective buried trench gates are formed only in the p-body, which is far from the bottom drain, as an effect, this helps weaken the dependence of gate-to-drain capacitance (Cgd) on cell pitch.

There are several sections in this work. The distinctive features, performance, and process flow of the suggested device are highlighted in Section 2, which shows the device construction with different channel materials and its comparison with the conventional power MOSFET. Section 3 introduces the SiGe triple gate MOSFET and its optimization. The outcome of the extracted 2-D ATLAS numerical simulation of the Si/SiGe/GeSn and traditional device, backed by analysis and debate, is displayed in Section 4 with the introduction of GeSn triple gate MOSFET. This paper is concluded in Section 5.

## 2. Device Structure, Performance, and Process Flow

Our proposed device is a selective buried triple gate power MOSFET consisting of a single vertical trench gate and two lateral selective buried gates, oriented orthogonally over the n-type drift region, as shown in Figure 1a below. In the figure, we represent the different types of channel material in the p-body by 1, 2, and 3 as Si, SiGe, and GeSn, respectively. Also shown is the direction of the current flow from the top to the bottom drain in the ON-state, in Figure 1b. Additionally, we used Silvaco ATLAS 2D [10] to evaluate our device’s improved performance with a simple conventional trench MOSFET (CTPMOS) derived from [11]. Additionally, Table 1 displays the optimized structure dimensions that were employed in our simulation work.

Based on the selective buried gate with trench gate technology, this research presents a high-performance silicon triple gate power MOSFET structure. The impact ionization model (IMPACT SELB), field-dependent mobility (FLDMOB), Auger recombination (AUGER) model, concentration-dependent mobility (CONMOB), concentration-dependent lifetime model (CONSRH), and bandgap narrowing (BGN) are among the models that are used to compare the results with the CTPMOS device using Silvaco Atlas V5.19.20. R.

For the first time, the suggested new device features remarkable orthogonally oriented laterally and vertically aligned selective buried polysilicon gates. According to our simulation results below, the parasitic n-p-n BJT transistor’s negative effects, which are present in the typical trench MOSFET, are greatly reduced in the suggested device. Hence, a good improvement in the Qgd-Ron.sp and Ron.sp-BV trade-off in the proposed device, helping in discarding any need to increase the area of the MOSFET unit cell and scaling down the device without degradation in device property, which was lacking in earlier power devices.

Figure 2a,b shows the input and output characteristics of the proposed device selective buried triple gate device. The proposed device is compared with the conventional MOSFET device taken from [11]. The simulation output characteristics of the proposed device outperform the conventional MOSFET, with output taken at various Vds and gate overdrive voltages. When the proposed triple gate MOSFET is in its on-state, current flows laterally in the channel on the selective buried gate and along the vertical trench sidewalls, simultaneously forming multiple conducting channels. Very high on-current and very low Ron.sp. are the outcomes of this phenomenon, which causes a high and uniform distribution of the on-state current density from the surface to the bottom. Figure 3 shows the Si triple gate and conventional trench gate device [11] OFF state BV characteristics. Predicting the exact measurement of the n+ substrate depth of the conventional device in ref [11] is difficult, leading to a slight difference in our extracted breakdown voltage value, as mentioned in ref [11]. Moreover, our proposed device has a higher breakdown voltage, which is attributable to the following fact, explained here. A critical component of our proposed device’s operation is the junction that forms at the p-body/n-drift contact. However, the existing channel doping will expand the depletion area farther inside the channel region, which may upset its breakdown voltage, if the channel region is in direct contact with the drift region, as is the case in the conventional trench gate MOSFET structure. Furthermore, in reverse-biased conditions, where this depletion zone expands and merges with the source region in the conventional MOSFET structure, an even more hazardous scenario arises. At lower reverse voltages, this leads to an early device failure. There is also a considerable chance of parasitic n-p-n activation.

In addition, the proposed device’s source–body and drift–body interface junctions are orientated differently, so current channel doping fluctuations will not cause them to merge, leading to a sharp and improved breakdown performance. Because the source body and drift body connectors are pointing in separate directions, the parasitic n-p-n effect is lessened, improving the performance of our proposed device. Our proposed triple gate power MOSFET outperforms the conventional MOSFET, giving 78.8 V and 64 V, respectively.

Device optimization is one of the assessing measures for the best performance of a device in terms of its output. We have optimized our device by changing the dimensions of the lateral and vertical trench gates. Figure 4 shows the effect on electron mobility with a change in vertical trench gate depth, extracted at x = 1.5 µm vertical cutline. We increased the vertical trench gate depth from 0.45 µm to 1.5 µm. In the graph, the maximum electron mobility was achieved at 0.45 µm. This gives us the high value of electron mobility at the minimum value of the vertical trench gate depth, resulting in the optimized trench gate depth = 0.45 µm. From the graph, we note that the maximum value of the electron mobility is achieved when the ON-current gathers, emerging from lateral and vertical inversion layers, thereby showing the super-controlling of the channel by lateral and vertical gates. Furthermore, to highlight the influence of vertical dimension (i.e., depth) on electron mobility, we note that, as the vertical depth (y) increases, an electron moves deeper inside the drift region (i.e., movement of an electron from the vertical top to bottom), toward the drain terminal, the electron experiences many changing phenomena like high drift doping concentration (i.e., 1 × 10^19^/cm^3^), high electric field (since the drain terminal is at high potential), and high scattering (due to collision), which cause the electron drift velocity to reach a saturation value (as seen in the graph; also, reductions in electron mobility). Moreover, the two optimized lateral selective buried gates dimension was taken at 0.65 µm, as discussed in [12].

The proposed selective buried triple gate device’s possible process flow is depicted in Figure 5. According to previous reports in the literature, the fabrication technologies are well-optimized based on the prevalent fabrication schemes and the device dimension values. As illustrated in Figure 5A,B, the first stage is the accurate formation of a lightly doped n-region epitaxial growth on an essentially n-n-doped substrate [13]. The oxidation technique for oxide layer deposition (Figure 5C) is then selected, followed by n+ Poly-Si gate deposition on the oxide, with selective etching stages to characterize the trench (Figure 5D,E), and thermal oxidation (Figure 5F [14,15,16,17,18]) under controlled pressure and temperature. Additionally, the upper part of the suggested device is made of epitaxial Si that has been further developed via a seed window (Figure 5G) [19]. Additionally, as illustrated in Figure 5H,I [20,21], the same process is followed to construct a trench gate (vertical) as it does for the two lateral gates. As seen in Figure 5J, ion implantation is next used to construct n+, p+, and p-body areas. To create connections for the source, drain, and gate terminals, metallization comes next. The proposed device process flow options shown above are comparable to those of the traditional method.

## 3. Si_(1−x)_Ge_x_ Triple Gate MOSFET

Due to advantages, including greater mobility than Si, a lower bandgap than Si, Si/SiGe heterostructures, and more, the SiGe material has just been available for the production of sophisticated devices.

The earliest known publication on SiGe discussed the magnetoresistance of silicon germanium alloys [8]. The first patent for the bipolar transistor, which is exactly where SiGe devices were originally mentioned, included a description of the physics of the SiGe base HBT concept in the 1950s [22]. In 1975, Erich Kasper at the AEG Research Centre in Ulm, Germany, demonstrated that epitaxial growth of SiGe heterostructures was required for such a transistor using molecular beam epitaxy (MBE). Since then, the sector has grown significantly, with emerging growth technology dominating the 1980s, HBT advancements dominating the 1990s, and Strained-Si CMOS dominating the early 2000s.

The carrier mobility in the channel region would be impacted by the Ge mole fraction of Si_(1−x)_Ge_x_ for this proposed device configuration, and the ON-state resistance could be altered. The Si_(1−x)_Ge_x_ channel region’s ON-state resistance is plotted against the Ge mole fraction in Figure 6. A lower ON-state resistance might be achieved with a greater Ge mole percentage since it would increase carrier mobility in the channel region. This results in a particular ON-state resistance of less than 0.358 (mΩ·cm^2^) for a Ge mole fraction of 0.2. A particular ON-state resistance of 1.22 (mΩ·cm^2^) is produced for the conventional Si-channel MOSFET [11] device at the same cell pitch. This gives us optimized parameters of the mole fraction (x = 0.2). Therefore, the SiGe channel is one of the good candidates to further increase the channel performance. To be noted here, this is the third fold optimization (after vertical and lateral trench gate optimization) of our proposed selective buried triple gate vertical power MOSFET. To begin with, both the lateral and vertical gate dimensions were optimized (L = 0.65 µm, T = 0.45 µm) for best output. In addition, further, the Ge mole fraction “x” is optimized, obtaining x = 0.2 for the lowest Ron.sp and high BV. All simulations are being performed using these optimized parameters. Further, in our work, we have shown the performance of the proposed device using Si_(1−x)_Ge_x_ as a channel material over Si, by comparing it with different channel materials. All simulations are extracted at the optimized device parameters in this paper. Furthermore, to examine the electrical properties with matched threshold levels and doping, we also simulated the suggested device. Though the on-current of the Si channel-based proposed device is deceased, it is still higher than the conventional power MOSFET [11].

## 4. Ge_(1−x)_Sn_x_ Triple Gate MOSFET

Since GeSn has significantly greater electron and hole mobility than Si, Sn, the element in group IV next to Ge, has attracted a lot of attention lately. Materials from group IV alloys, including SiGe, Ge [8], and germanium-tin (GeSn) [9], have received a lot of interest lately because of their increased carrier mobility and excellent compatibility with the conventional Si manufacturing technique. To improve the channel mobilities, high mobility channel materials like GeSn have been studied further. These materials include fin, nanosheet, and nanowire (NW) channel topologies.

### 4.1. Performance of GeSn Triple Gate Power MOSFET with Other Channel Material

The triple gate selective buried MOSFET’s performance is extracted with GeSn as channel material and is compared with other channel materials including Si and SiGe. Figure 7 represents the input characteristics of the proposed triple gate. The GeSn triple gate outperforms the Si and SiGe channel materials. A very high channel current is produced in the device with GeSn as channel material.

Figure 8 shows the output current of the proposed device, with GeSn channel material giving a very high current. This increase in drain current is caused by both the carrier mobility enhancement and transconductance in our proposed device, and, moreover, the superb compatibility of GeSn with the conventional Si. From the above two characteristics, we can infer that, using GeSn as the channel materials, the proposed device has the least specific ON-resistance compared to SiGe and Si counterparts.

The breakdown voltage was also extracted at Vgs = 0 V. Though the voltage handling breakdown voltage of GeSn decreased in comparison to Si used as channel material, it is higher than SiGe, as shown in Figure 9. Furthermore, compared to the conventional MOSFET, the breakdown characteristics show a 12.5% rise in the voltage handling capacity of the proposed GeSn triple gate MOSFET, indicating a substantial enhancement in the Ron.sp-BV tradeoff.

### 4.2. Device Sensitivity, Mobility, and Gate Charge Analysis

The MOSFET’s sensitivity to changes in input voltage is shown by the transconductance value. In small-signal applications, it establishes the MOSFET’s amplification capabilities. More amplification and better linearity are made possible by a higher transconductance value, which qualifies MOSFETs for use in RF circuits and audio amplifiers. For this, the three devices’ transconductance (gm) was extracted using three different channel materials, as shown in Figure 10, below. Out of the three, using GeSn gives the most significant output, thus, indicating the device’s best sensitivity using GeSn as channel materials.

Also, the primary factor influencing the accuracy of the drain current calculation findings utilizing 2D and 3D simulation tools is the channel electron mobility. Hence, we further analyzed the device performance, and the electron mobility in the device was extracted for the three devices. Figure 11 predicts the scenario of electron mobility in the three devices using different channel materials. This is due to the fact that our proposed device has a high carrier confinement effect due to the Si layer above and below the GeSn channel. This causes the electrons to reside mainly in the GeSn layer, hence, the channel mobility is greatly enhanced, compared to conventional Si MOSFET. GeSn tops among the three, giving a maximum electron mobility of about 880 cm^2^/V·s. We believe this value to be very high until now in the triple gate MOSFET domain.

The gate voltage versus transient time characteristic for the Si/SiGe/GeSn channel triple gate MOSFET device is obtained using the circuit [23], as seen in Figure 12. A closer look at the gate charge (Qgd) simulation curve is shown in Figure 13. Gate-to-drain charge (Qgd) is directly related to power MOSFET switching speed and loss [23]. Our suggested device gate charge (Qgd) is improved compared to other published material and has a lower Qgd. In the suggested Si/SiGe/GeSn channel triple gate MOSFET, the low value of Qgd indicates the reduced gate-drain overlap and the good shielding effect, as seen in Figure 12 and Figure 13.

To charge the gate for extracting Qgd during simulation, we connected 10^4^ cells in parallel with a constant current of 10 μA. In comparison to the CTPMOS structure, the suggested device’s gate charge is just 3.4 pC, indicating a 94.89% reduction in both gate charge and parasitic impact, as shown in Figure 13.

Furthermore, it is not worthwhile to examine each of the three important performance metrics separately (BV, Qgd, and Ron.sp). As a result, the precise device’s performance—the tradeoff between the three parameters—is eventually determined by the two well-known Baliga’s Figures of Merits (FOM1, FOM2). The most effective method for evaluating the various device kinds and assessing their performance is to compare them in terms of tradeoffs [22]. Table 2 shows that, in comparison to the CTPMOS device, our suggested device much improves the retrieved Baliga’s figures-of-merit, FOM1 and FOM2.

The ideal silicon-limit line and other devices that have been reported in the literature within the range of 0–110 V power MOSFET are also used to benchmark the Balliga’s FOM of the three proposed (Si/SiGe/GeSn) devices, as shown in Figure 14. Additionally, the values of the specific ON-resistance (Ron.sp) and BV of the three suggested devices are also included. It is clear from the graph that the three suggested devices perform well in comparison to the other devices that have been documented in the literature. The graph lists the references for the reported structure along with the BFOM values that correspond to them.

Figure 15 and Figure 16 are the benchmarking of our three devices with GeSn, SiGe, and Si as channel materials against various fabricated and reported literature using GeSn, SiGe, and Si transistors. As seen in Figure 15, the three proposed devices’ ON-currents (Ion) are quite high compared to other fabricated and reported literature, except for a few. Similarly, Figure 16 shows the benchmarking between the electron mobility, comparing our proposed three devices and various fabricated and reported literature. Our work is the first to combine orthogonally orientated selective buried trench gates using Si/SiGe/GeSn to produce a triple gate power MOSFET, as far as we are aware. For a broader understanding, we are, therefore, inspired to compare our device with several types of power transistors. On the ion and electron mobility scale, our suggested device exhibits a noticeably lower Ron.sp and higher I_ON_^max^ (among Si) than many other devices, except for a small number of pieces of published literature (based on GaN and SiC). Additionally, a bar graph in Figure 17 illustrates the % improvement of several parameters of the suggested GeSn-based triple gate device in comparison to the CTPMOS device for greater consideration. The suggested device works well in all trench-gate power MOSFET governing parameters, according to our 2D ATLAS simulated results, at the cost of a few simple additional process steps.

## 5. Conclusions

A novel state-of-the-art selective buried Si/SiGe/GeSn-based orthogonally oriented vertical triple-gate power MOSFET is designed and studied. To further enhance the functionality of CMOS technology, the Si, SiGe, and GeSn channel MOSFETs have been investigated in this work. Many inversion layer channels are incorporated into the vertical and lateral sidewalls of the suggested device. For the first time, we suggest combining trench and buried gates oriented orthogonally to create a triple gate power MOSFET. In terms of extremely high ON-current and low specific ON-resistance, the suggested device performs better. The unique design of our proposed triple gate gives a very high channel mobility of 880 cm^2^/V·s, which we believe to be significant in the triple gate power MOSFET domain. According to subsequent results, our three-time optimized proposed triple gate device has a high switching and speed capacity, improving Balliga’s FOM1 and FOM2 by 411.61% and 98.704%, respectively. It uses various high-mobility channel materials and has an ultra-low specific ON-resistance of 0.31 mΩ·cm^2^, making it suitable for future CMOS devices. When the Ge_(1−x)_Sn_x_ channel has a cell pitch of 3.0 µm, the most optimal device lateral selective buried and vertical trench gate dimensional parameters are 0.65 µm and 0.45 µm at a mole fraction of x = 0.2, respectively. Furthermore, using the circuit implementation, the suggested device has numerous channels produced in the two p-body areas above the two lateral gates and two sidewalls of the vertical trench gate, all at once, far from the drain, providing a very low value of the gate-to-drain charge. The suggested triple-gate MOSFET expands on the current multi-gate strategies for lowering conduction and switching loss in semiconductor power devices. While this study contributes valuable insights into the effects of enhancing channel mobility, several areas warrant further exploration. One avenue for future research is to conduct a study of the strain produced between Si-GeSn-Si layers and its effect on device performance. Other areas to be explored are the study of the temperature range for each technological process, for evaluating the possibility of polycrystalline Si restructuring during any of the technological operations. This would allow for a more comprehensive understanding of how these parameters impact device performance.

## Figures and Tables

**Figure 1 micromachines-16-00452-f001:**
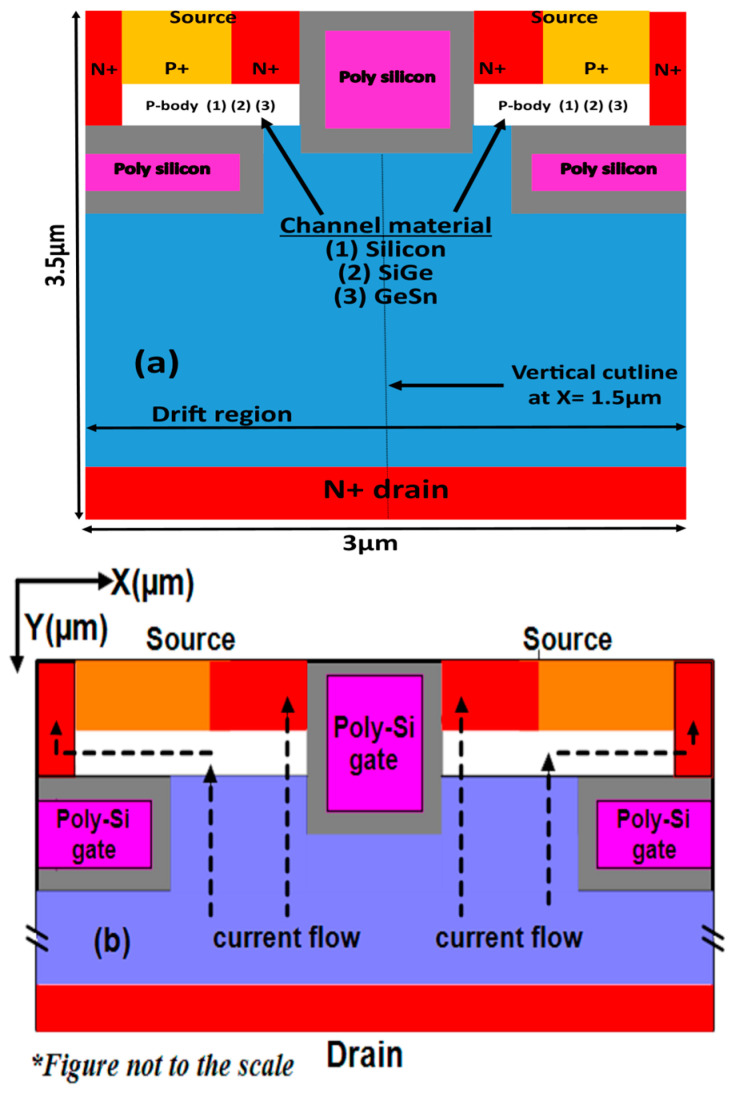
(**a**) Two-dimensional cross-sectional description of the proposed triple gate vertical trench power MOSFET, showing device vertical cutline. (**b**) Current flowing in the proposed device during ON-state.

**Figure 2 micromachines-16-00452-f002:**
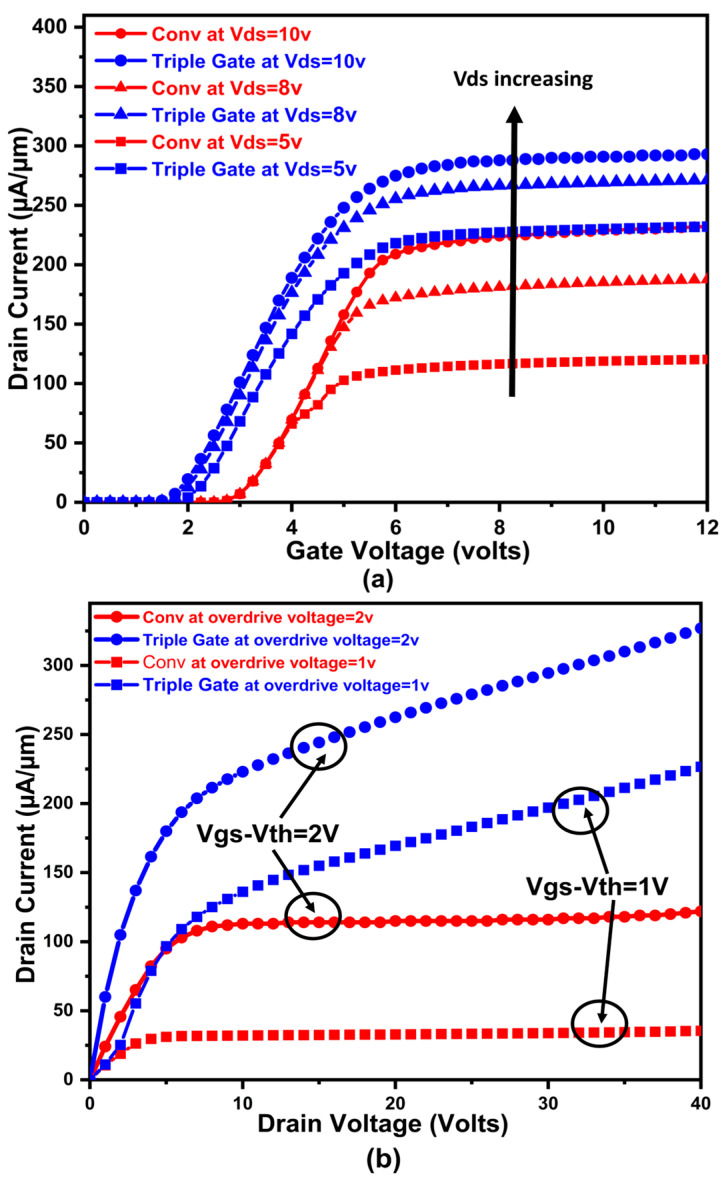
(**a**) Proposed device transfer characteristics with various Vds. (**b**) Proposed device output characteristics with gate overdrive voltages.

**Figure 3 micromachines-16-00452-f003:**
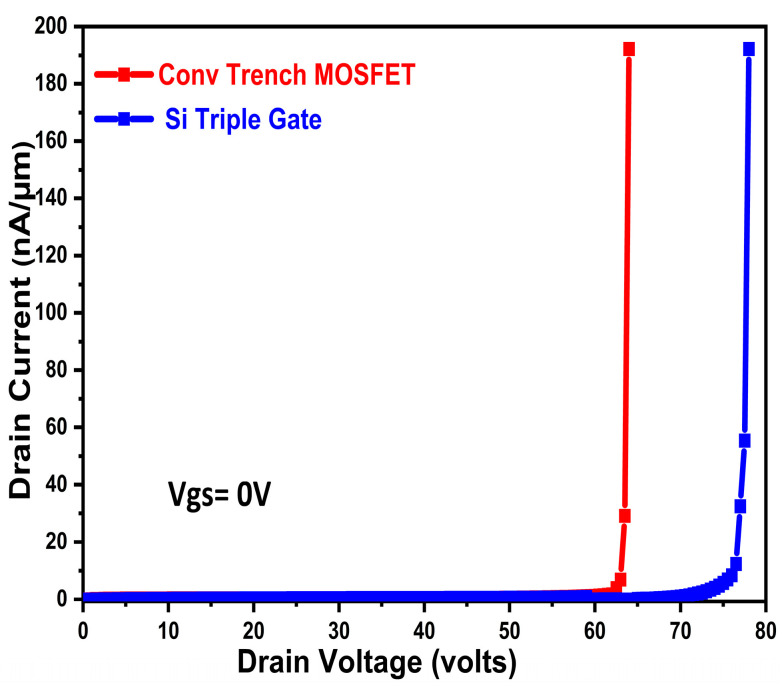
Si triple gate and conventional triple gate device OFF state BV characteristics.

**Figure 4 micromachines-16-00452-f004:**
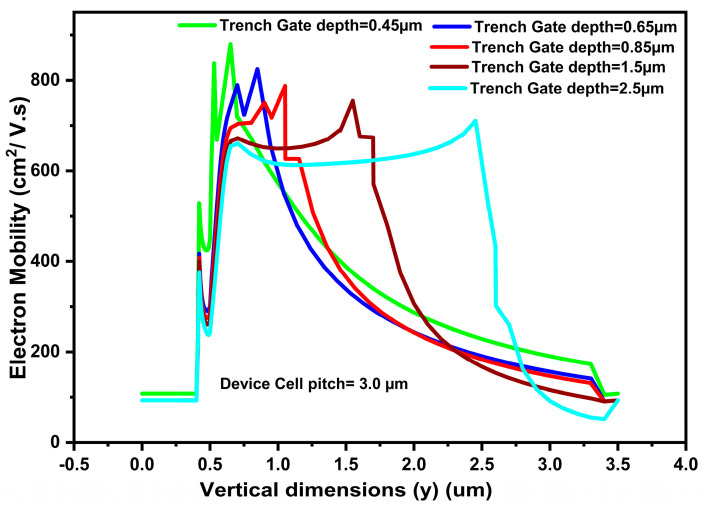
Variation in device electron mobility with change in trench gate depth in vertical (y) position at device cell pitch = 3.0 µm.

**Figure 5 micromachines-16-00452-f005:**
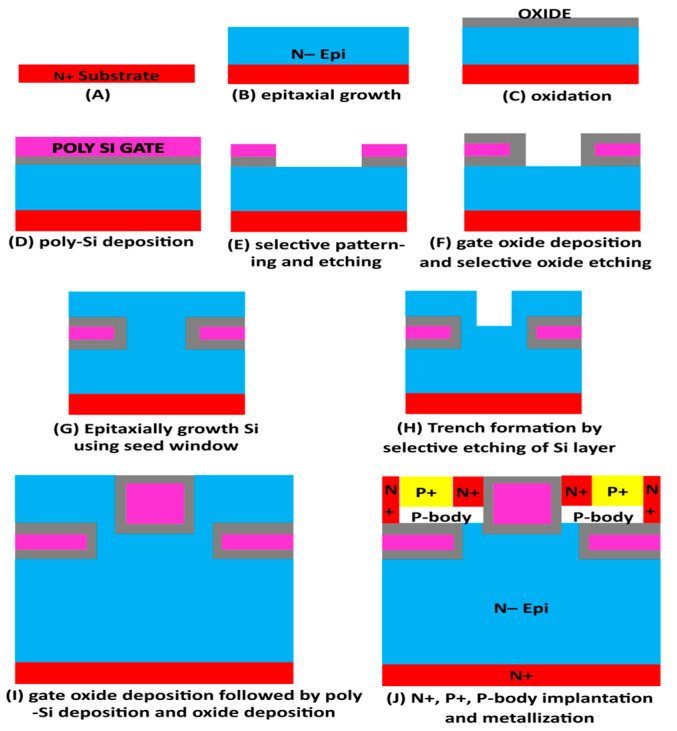
Proposed device fabrication process flow chart/diagram.

**Figure 6 micromachines-16-00452-f006:**
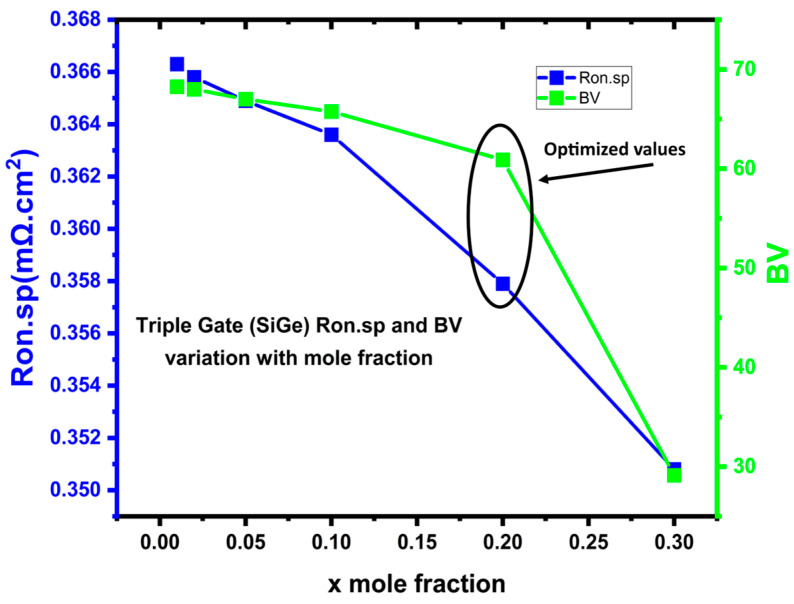
Breakdown (off state) voltage and ON-resistance variation with mole fraction Si_(1−x)_Ge_x_ triple gate power MOSFET.

**Figure 7 micromachines-16-00452-f007:**
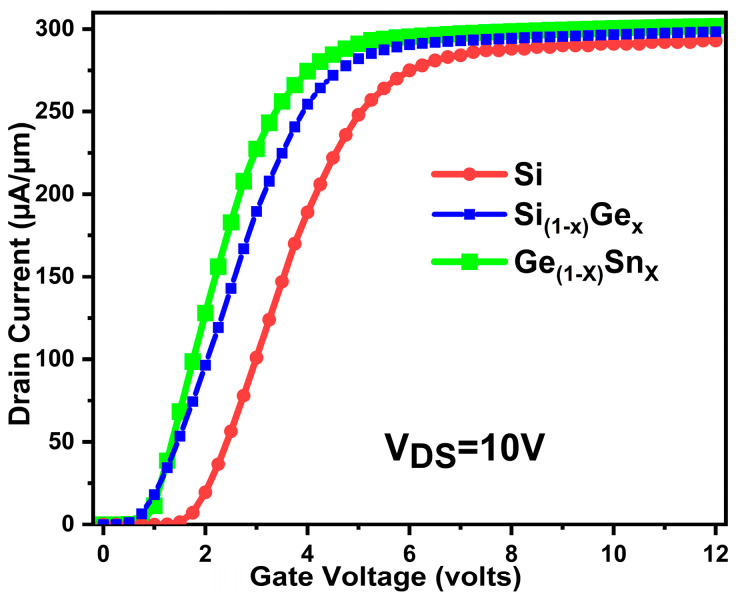
Si/SiGe/GeSn channel triple gate device input characteristics.

**Figure 8 micromachines-16-00452-f008:**
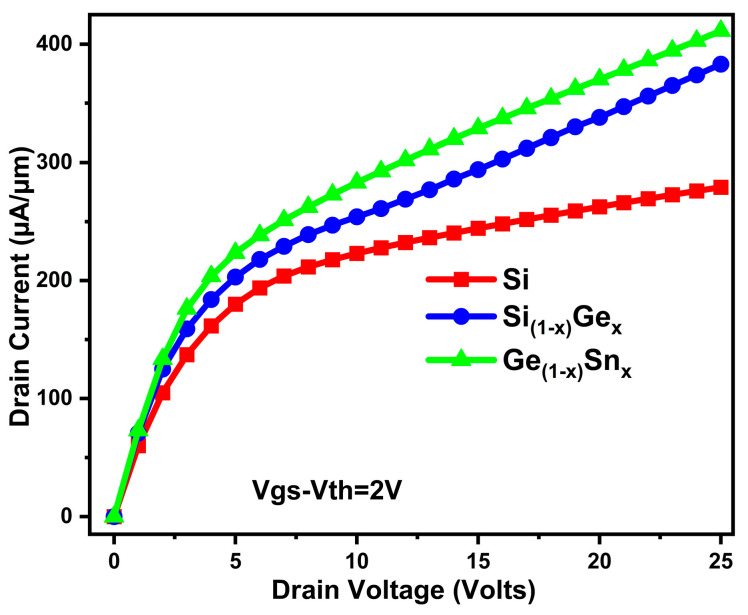
Si/SiGe/GeSn channel triple gate device output characteristics.

**Figure 9 micromachines-16-00452-f009:**
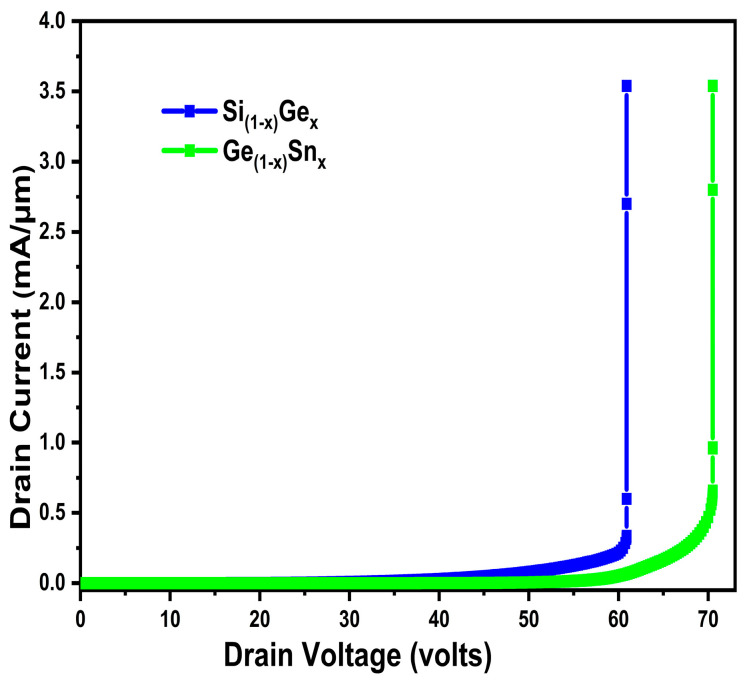
SiGe/GeSn channel triple gate device OFF state BV characteristics.

**Figure 10 micromachines-16-00452-f010:**
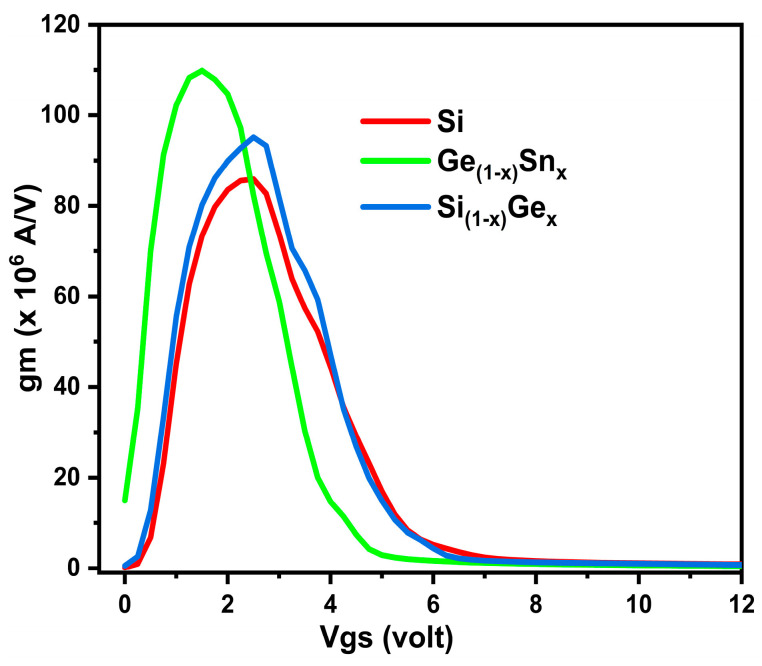
The three suggested devices (Si/SiGe/GeSn) gm variations with Vgs.

**Figure 11 micromachines-16-00452-f011:**
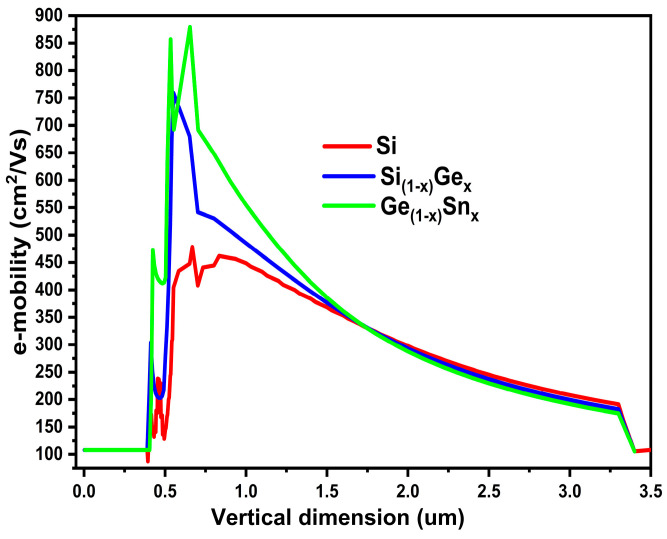
Various device region e-mobility distribution for the three devices, in the vertical (y) position, at device cell pitch = 3.0 µm.

**Figure 12 micromachines-16-00452-f012:**
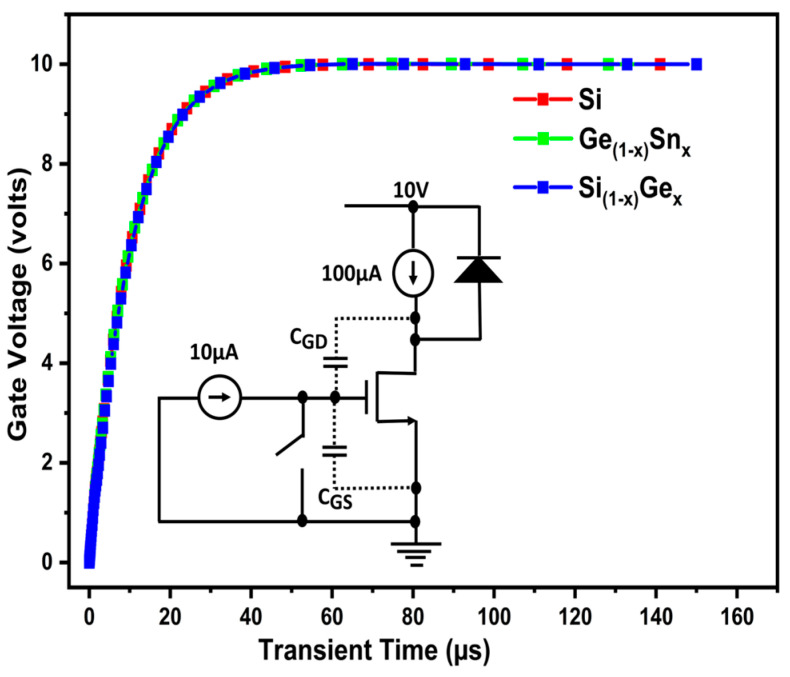
Gate charge simulation output with circuit [23].

**Figure 13 micromachines-16-00452-f013:**
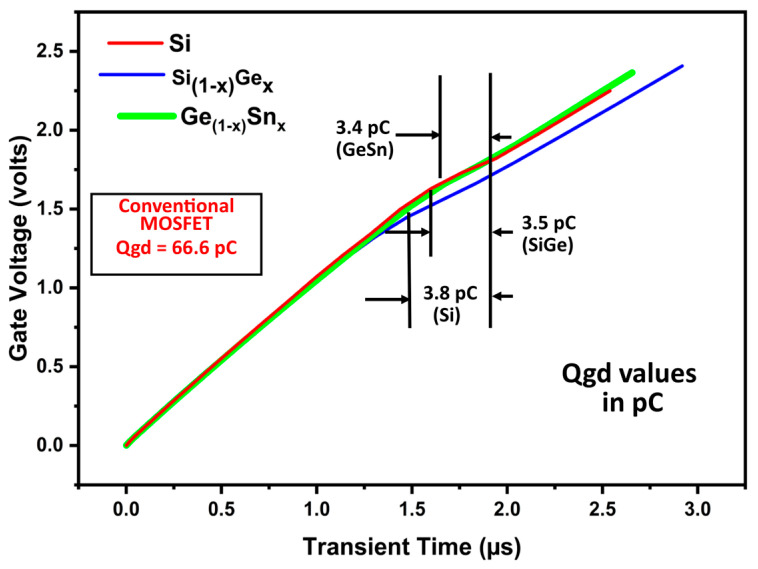
Closer view of the gate charge simulation output using circuit [23]. The conventional MOSFET [11] Qgd (in pC) is also shown inside the image.

**Figure 14 micromachines-16-00452-f014:**
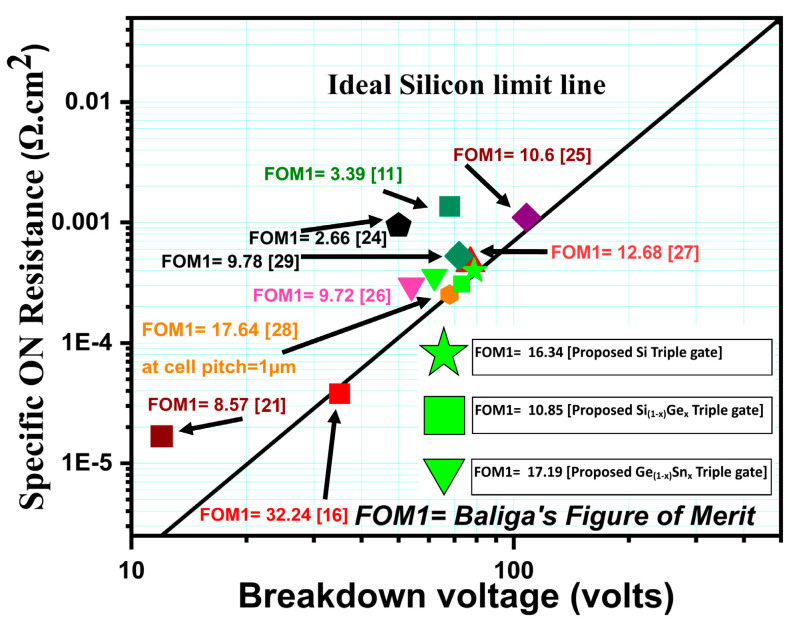
The three suggested devices (Si/SiGe/GeSn) have specific-ON-resistance (Ron.sp) vs. BV with the silicon limit line (ideal) against different vertical and lateral power transistors, with corresponding values of Baliga’s FOM1. References [11,16,21,24,25,26,27,28,29] are mentioned in image.

**Figure 15 micromachines-16-00452-f015:**
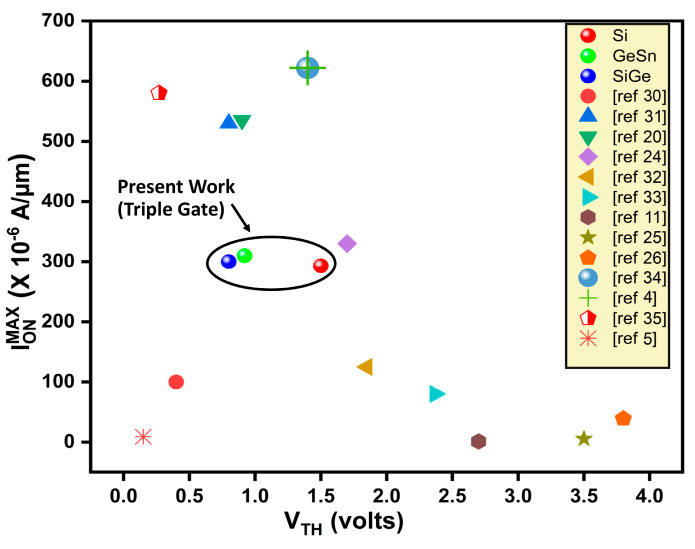
Benchmarking of I_ON_^max^-V_TH_ of the proposed device against various Si, GaN vertical, and lateral power transistors (Si, SiGe, GeSn), extracted at same units. References [4,5,11,20,24,25,26,30,31,32,33,34,35] are mentioned in image.

**Figure 16 micromachines-16-00452-f016:**
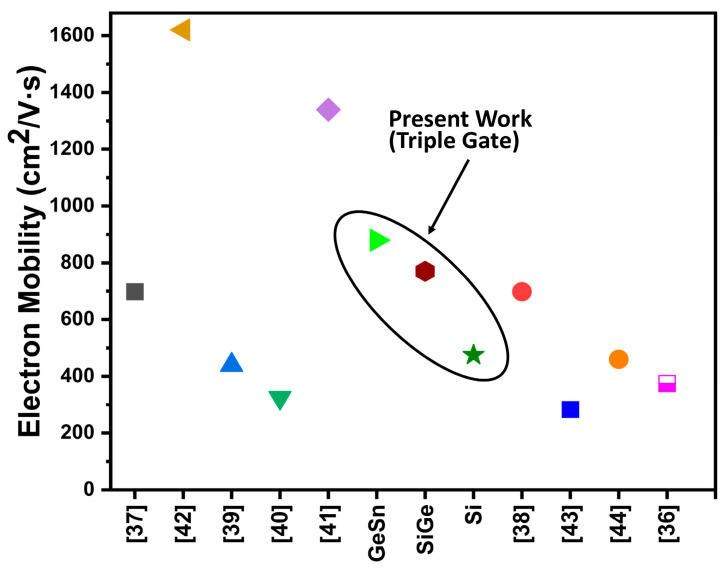
Benchmarking of electron mobility of the proposed device with GeSn, SiGe, and Si as channel length against various fabricated and reported literature using GeSn, SiGe, and Si transistors. References [36,37,38,39,40,41,42,43,44] are mentioned in image.

**Figure 17 micromachines-16-00452-f017:**
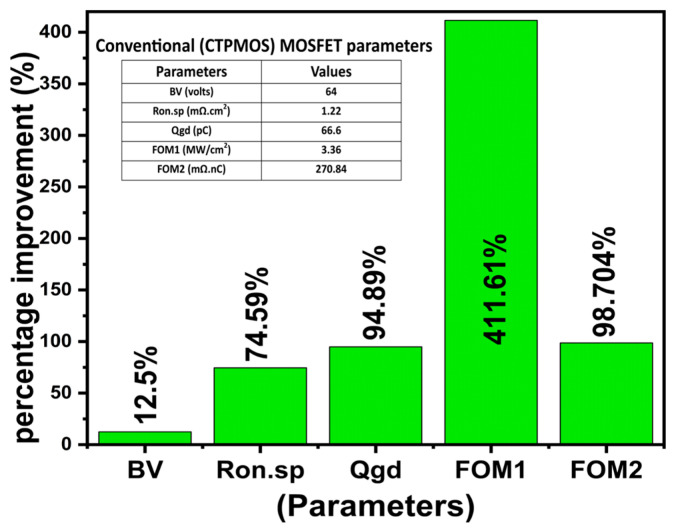
Percentage improvement illustrating the enhanced improvement in Ron.sp, FOM1, Qgd, BV, and FOM2 for the proposed triple gate using the GeSn device over the conventional (CTPMOS) device [11].

**Table 1 micromachines-16-00452-t001:** Proposed and conventional device dimension and doping values.

Parameters	Conventional Trench Power MOSFET [11]	Proposed Vertical Triple Gate Device
Cell pitch	3 µm	3 µm
Trench width	1 µm	Lateral gate = 0.65 µm (each), Vertical gate = 1 µm
Trench depth	1.3 µm	Lateral gate = 0.1 µm (each), Vertical gate = 0.45 µm
Gate oxide thickness	50 nm	50 nm
N+ source width along vertical trench	----	0.334 µm
Source doping	1 × 10^19^/cm^3^	1 × 10^19^/cm^3^
Source depth	0.2 µm	0.5 µm
N-drift region doping	5.6 × 10^14^/cm^3^	5 × 10^15^/cm^3^
p-body doping	1.7 × 10^17^/cm^3^	1.2 × 10^17^/cm^3^
P+ doping	2 × 10^19^/cm^3^	2 × 10^19^/cm^3^
N+ drain doping	1 × 10^19^/cm^3^	1 × 10^19^/cm^3^

**Table 2 micromachines-16-00452-t002:** Performance comparison of Baliga’s FOM for proposed triple gate mosfet with different channel materials.

Figure of Merit (FOM)	Proposed Triple Gate (Si Channel)	Proposed Triple Gate (SiGe Channel)	Proposed Triple Gate (GeSn Channel)
FOM1 (BV^2^/Ron)	16.34 (MW/cm^2^)	10.85 (MW/cm^2^)	17.19 (MW/cm^2^)
FOM2 (Ron. Qgd)	4.5 (mΩ·nC)	4.08 (mΩ·nC)	3.51 (mΩ·nC)

## Data Availability

The original contributions presented in this study are included in the article. Further inquiries can be directed to the corresponding author.

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
