# Peer review of "Investigation of Channel Mobility Enhancement Techniques Using Si/SiGe/GeSn Materials in Orthogonally Oriented Selective Buried Triple Gate Vertical Power MOSFET: Design and Performance Analysis"

_micromachines, 2025, doi:10.3390/mi16040452_

Round 1
Reviewer 1 Report (Previous Reviewer 1)
Comments and Suggestions for Authors
The revised version addressed the review.
Author Response
Dear Reviewer
Thank you so much for your wise review, all you suggestions are taken seriously andare incorporated in the present version of manuscript.
Reviewer 2 Report (Previous Reviewer 2)
Comments and Suggestions for Authors
The authors have improved the paper significantly and clarified most of the points. However the paper is not fully free of ambiguities yet. Some of the issues have remained unresolved.
Considering the previous report:
1) The authors use the MOSFET device discussed in Ref.10 as a reference one... However, the exact device configuration and thicknesses of the layers are different...
The authors have added some useful discussion clarifying the comparison. However it is still unclear which exact parameters provide the performance improvement. The corresponding enhanced discussion is highly desirable.
Also the parameter N+source width along the vertical trench is not presented in ref.10.
5) Lines 49-50. When discussing the characteristics, the authors mention the ON-current (which is a physical property), a tradeoff (which has no qualitative indicator) and coupling for which the indicator might be introduced. I believe this is incorrect to discuss the above characteristics in one context.
It is much more convincing to discuss either quantitative (measurable) or qualitative (unmeasurable) parameters within one argument. Since tradeoff has a quantitative characteristic (FOM1) it is better to be provided alongside the ON-current.
8) Lines 143-146 the sentence is very hard to understand. It is not clear whether increase or decrease of gate dimension should improve the mobility.
The discussion at Lines 147-151 still seems incomplete. The authors discuss the dependence of mobility with trench gate dimension. However the graph represent a set of dependences of mobility on vertical dimensions. These dependences are also to be discussed at least briefly.
11) Figure 5 - The fabrication process proposed involves high temperature operations (epitaxy, post-implantation anneal) that would lead to restructuring of polycrystalline silicon.
Reply: Sir/Madam, we have taken utmost care in considering all factors raised by you. Moreover, the proposed device process flow options shown are comparable to those of the traditional method. According to previous reports in the literature (references in manusript), the fabrication technologies are well optimized based on the prevalent fabrication schemes and the device dimension values, considering all aspects.
Comment: Anyway I recommend to provide the temperature range for each of the technological process so it will be possible to evaluate the possibility of polycrystalline Si restructuring during any of the technological operations.
12) The calculation do not take into account the strain caused by the difference between the lattice parameters of Si and GeSi (GeSn). The strain relaxation would lead to defect formation which in turn would decrease the electron mobility.
I would agree with the argumentation of the authors. However I recommend to mention in the text of the paper that discussion of the strain in the layer is to be done in the future work.
The revised paper makes much better impression, so I believe it can be published as soon the authors provide comments on the remaining issues.
Author Response
Dear Reviewers(s)
Thank you very much for providing the opportunity for further improvisations.
Please refer to attached file.

Round 2
Reviewer 2 Report (Previous Reviewer 2)
Comments and Suggestions for Authors
The authors have introduced some improvements that were recommended. However some of the issues remain unaddressed.
1) The authors use the MOSFET device discussed in Ref.10 as a reference one... However, the exact device configuration and thicknesses of the layers are different...
The authors have added some useful discussion clarifying the comparison. However, it is still unclear which exact parameters provide the performance improvement. The corresponding enhanced discussion is highly desirable.
Reply: Sir/Madam, we have taken the conventional structure design from ref [10]...
Comment: I find the explicit comment provided by the authors very useful. I recommend to introduce these consideration into the discussion section.
8) The discussion at Lines 147-151 still seems incomplete. The authors discuss the dependence of mobility with trench gate dimension. However, the graph represents a set of dependences of mobility on vertical dimensions. These dependences are also to be discussed at least briefly.
Reply: We have modified the sentence as per your kind views and updated in the manuscript.
Comment: Still the dependence of the Electron mobility on the Vertical dimensions (y) is not provided. The maximum mobility is revealed at y=0.25 mkm. What is the reason for this position of the maximum. The influence of Vertical dimensions on the electron mobility is to be discussed.
The two questions above are the major unaddressed issues. I recommend to discuss them in the paper.
Author Response
Thank you so much for your valuable inputs and providing chance for further improvisations. The detailed response are attached herewith.

This manuscript is a resubmission of an earlier submission. The following is a list of the peer review reports and author responses from that submission.
Round 1
Reviewer 1 Report
Comments and Suggestions for Authors
This paper investigated channel-mobility enhancers of SiGe/GeSn materials in buried triple-gate vertical power MOSFET and revealed improved switching and lowered specific ON-resistance. However, various simulated transfer characteristics in Figs.2 and 7 shows low output currents below 1mA/um even though epitaxial quality and device integration were not considered yet. Moreover, it is difficult to compare the claimed channel-mobility of 880 cm2/V.s obtained in such sub-micro devices with those measured at the highly challenging logic nano-FETs. Moreover, Fig. 3 gives the BV comparison but it seems not record-high data (conventional trench devices) reported.
Low output currents even though the influences of epitaxial quality and device integration were not considered need clarification/explanation.
To study the impact of claimed channel-mobility of 880 cm2/V.s in such sub-micro devices on output currents/BV/speed in terms of device structures is important.
Comparisons among present devices and representative devices announced is suggestive.
Author Response
Dear Editor and Reviewers
We are extremely thankful for reviewing our manuscripts. Your invaluable suggestions are appreciated. We have incorporated all the suggestions to improvise the manuscript. Point wise response to your recommendations are attached here.

Reviewer 2 Report
Comments and Suggestions for Authors
The paper is devoted to analysis of the nely designed MOSFET scheme (Orthogonally Oriented Selective Buried Triple gate vertical power MOSFET) that includes two buried poly-Si gates. The authors have performed the calculations using Silvaco ATLAS 2D. Comparison of the designed device with the more or less conventional MOSFET has demonstrated better performance of the designed scheme. The use of Ge-Si solid solution or GeSn has allowed further improving the performance.
The paper includes some elements of the novelty, however the results presentation is very much lacking the details and explanation of the physics. The authors focus the calculation of the performance characteristics paying much fewer attention to analyzing the physical performance.
1) The authors use the MOSFET device discussed in Ref.10 as a reference one. The device design is indeed somehow close to the investigated one. However the exact device configuration and thicknesses of the layers are different. The exact performance characteristics are strongly dependent both on the thicknesses of the layers and on lateral dimensions. The comparison of the performance parameters requires more detailed discussion of interrelations between the layers properties and device characteristics.
2) The device description is not sufficient. The authors of Ref.10 have plotted the the distribution of potential in the trench-channel region. Also the current flo direction would be very useful for understanding the device performance.
3) Line 37 "mostly because of the device's large active area" the proper reference is required.
4) Lines 40-41. The "sudden changes in the electric field between the two gates" referred to Ref.10 is to be specified in more details.
5) Lines 49-50. When discussing the characteristics the authors mention the ON-current (which is a physical property), a tradeoff (which has no qualitative indicator) and coupling for which the indicator might be introduces. I believe this is incorrect to discuss the above characteristics in one context.
6) Lines 71-74. The paragraph at lines 71-74 discusses the global problem of the MOSFET devices, so it is better to be moved to the introduction section. At this section of the paper the specific structures are already discussed.
7) Lines 79-81. The sentence actually claims that the multiple gates result in high electron mobility. Such thesis is very controversial.
8) Lines 143-146 the sentence is very hard to understand. It is not clear whether increase or decrease of gate dimension should improve the mobility.
9) Line 154 "good thermal oxidation". The "good" term introduces ambiguity since it cannot be evaluated numerically.
10) Figure 4 - The vertical cutline is to be plotted at the device scheme to clarify the understanding.
11) Figure 5 - The fabrication process proposed involves high temperature operations (epitaxy, post-implantation anneal) that would lead to restructuring of polycrystalline silicon.
12) The calculation do not take into account the strain caused by the difference between the lattice parameters of Si and GeSi (GeSn). The strain relaxation would lead to defect formation which in turn would decrease the electron mobility.
As the final remark the paper requires significant restructuring and rethinking without which it cannot be recommended for the publication.
Comments on the Quality of English Language1) Lines 25-29. The sentence is not consistent.
2) Lines 42-43. The sentence is to be checked for the consistency.
3) Lines 64-65 the misprint in paragraph
4) Line 67 I believe it is better to use "despite" instead of "although".
5) Lines 143-144 The sentence is to be checked for the consistency.
5) Lines 85-86 Please avoid using the word "contribute" two times in the same sentence.
Author Response
Dear Editor and Reviewers
We are extremely thankful for reviewing our manuscripts. Your invaluable suggestions are appreciated. We have incorporated all the suggestions to improvise the manuscript. Point wise response to your recommendations are attached here.
Thank you so much for your insightful review.
